# MiR-155 Dysregulation Is Associated with the Augmentation of ROS/p53 Axis of Fibrosis in Thioacetamide-Induced Hepatotoxicity and Is Protected by Resveratrol

**DOI:** 10.3390/diagnostics12071762

**Published:** 2022-07-21

**Authors:** Amal F. Dawood, Suliman Al Humayed, Maha A. Momenah, Mohamed El-Sherbiny, Hend Ashour, Samaa S. Kamar, Asmaa M. ShamsEldeen, Mohamed A. Haidara, Bahjat Al-Ani, Hasnaa A. Ebrahim

**Affiliations:** 1Department of Basic Medical Sciences, College of Medicine, Princess Nourah bint Abdulrahman University, Riyadh 11671, Saudi Arabia; afdawood@pnu.edu.sa; 2Department of Internal Medicine, College of Medicine, King Khalid University, Abha 62529, Saudi Arabia; s_humayed@yahoo.com; 3Department of Biology, College of Science, Princess Nourah bint Abdulrahman University, Riyadh 11671, Saudi Arabia; mamomenah@pnu.edu.sa; 4Department of Basic Medical Sciences, College of Medicine, AlMaarefa University, P.O. Box 71666, Riyadh 11597, Saudi Arabia; msharbini@mcst.edu.sa; 5Department of Physiology, College of Medicine, King Khalid University, Abha 62529, Saudi Arabia; drhend_a_hassan@yahoo.com (H.A.); balani@kku.edu.sa (B.A.-A.); 6Department of Physiology, Kasr Al-Aini Faculty of Medicine, Cairo University, Cairo 12624, Egypt; dr_asmaashams82@cu.edu.eg (A.M.S.); haidaram@cu.edu.eg (M.A.H.); 7Department of Histology, Kasr Al-Aini Faculty of Medicine, Cairo University, Cairo 12624, Egypt; samaakamar@cu.edu.eg

**Keywords:** thioacetamide, miR-155, ROS/p53 axis, TIMP-1, α-SMA, liver fibrosis, resveratrol

## Abstract

Liver fibrosis is a hallmark of thioacetamide (TAA) intoxications. MicroRNAs (miRs), such as miR-155, have been implied in the pathogenesis of liver disease, and regulated by the antioxidant and anti-inflammatory compound resveratrol (RES). The link between reactive oxygen species (ROS), tumour suppressor p53 (p53), and liver fibrosis-during the pathogenesis of TAA-induced liver injury-associated with miR-155 dysregulation with and without RES incorporation has not been previously studied. Therefore, one group of rats received TAA injections of 200 mg/kg; twice a week at the beginning of week 3 for 8 weeks (TAA group; or model group), whereas the protective group was pretreated daily with RES suspension (20 mg/kg; orally) for the first two weeks and subsequently sustained on receiving both RES and TAA until being sacrificed at the 10th week. Liver injuries developed in the model group were confirmed by a significant (*p* < 0.0001) elevation of hepatic tissue levels of miR-155, ROS, p53, and the profibrogenic biomarkers: tissue inhibitor of metalloproteinases-1 and α-smooth muscle actin, as well as collagen deposition (fibrosis). All these parameters were significantly (*p* ≤ 0.0234) protected by resveratrol (RES + TAA). In addition, we observed a significant (*p* < 0.0001) correlation between ROS/p53 axis mediated liver fibrosis and miR-155. Thus, TAA intoxication induced miR-155 imbalance and ROS/p53-mediated liver fibrosis, with resveratrol, conversely displaying beneficial hepatic pleiotropic effects for a period of 10 weeks.

## 1. Introduction

Hepatotoxicity such as liver fibrosis and cirrhosis and hepatocarcinomas induced by the organosulfur compound thioacetamide (TAA), which was widely used in the industry, is now used as a model for these diseases in animals [1,2]. Liver disease is one of the primary causes of death due to organ failure and has become a global health burden, which sometimes requires liver transplantation as the only treatment option [3]. Liver fibrosis and cirrhosis are reported in mice and rats exposed to TAA for 6–10 weeks [2]. Pathological insults to the liver as toxins, metabolic diseases, alcohol abuse, autoimmune diseases, and viruses can cause liver fibrosis [4]. Therefore, preventing the development of hepatic disease to liver cirrhosis and eventually hepatic failure would be a wise therapeutic choice [5].

Liver injury induces the activation and then differentiation of the resident hepatic stellate cells (HSCs) into myofibroblast-like cells that produce the extracellular matrix in fibrosis, mainly fibrillar collagen [6]. Fibrillar collagen form the majority of the deposited collagen in human cirrhotic liver, and the intervention of the activation process of HSCs would be an important step to prevent the occurrence of liver fibrosis [7]. Interestingly, ROS is involved in the differentiation process of HSCs to myofibroblast-like cells [8,9]. ROS is also implicated in the development of metabolic liver disease and in chronic liver injuries induced by TAA [10]. Administration of antioxidants such as coffee and quercetin prevented TAA-induced hepatic injury [11]. Induction of the apoptosis biomarker p53 is reported to induce liver and cardiac fibrosis [12,13], in addition to the biomarker of profibrogenesis, a tissue inhibitor of metalloproteinases-1 (TIMP-1) that promotes liver fibrosis via the inhibition of the antifibrotic metalloproteinases upon the activation of HSCs [14]. On the other hand, miR-155 was reported to be involved in liver disease. For example, (i) liver tissue levels of miR-155 is elevated in patients with hepatitis C virus and declined in peripheral monocytes of patients responded to the antiviral therapy [15] and (ii) alcohol-induced steatosis, inflammation, and liver fibrosis were achieved via miR-155 upregulation, which was protected in miR-155 knockout mice [16]. Moreover, miR-155 was shown to induce the production of ROS in mesenchymal stem cells obtained from old mice [17].

Resveratrol is a polyphenolic natural compound highly abundant in the skin of red grapes [13]. It is widely used in research, since it exhibits beneficial pleiotropic effects, such as cardiovascular and kidney protection [18,19], prevents apoptosis and promotes cell survival [20], and inhibits the release of biomarkers of preeclampsia from human placenta and human umbilical vein endothelial cells [13,21]. Previous reports documented a beneficial role of resveratrol on the liver. For example, resveratrol (i) inhibits liver steatosis [22]; (ii) inhibits the proliferation of rat HSCs [23]; (iii) inhibits liver p53 gene expression in paracetamol-mediated acute hepatic injury in rats [24]; (iv) inhibits lipopolysacharride-induced inflammation and hepatic fibrosis in mice [25]; (v) suppresses cholestasis-induced liver injury and fibrosis [26]; and (vi) reduces miR-155 induced cardiac hypertrophy [27]. Therefore, these reports urged us to speculate that the modulation of miR-155 and activation of ROS/p53-mediated hepatic fibrosis by TAA, which could be prevented by resveratrol.

## 2. Materials and Methods

### 2.1. Experimental Design

All rat work was performed under project license number 20-0342 issued by the ethical committee at Princess Nourah bin Abdulrahman University. We followed the Guide for the Care and Use of Laboratory Animals published by NIH No. 85-23, revised 1996. Albino male rats (180–200 gm) were separated into three groups (n = 8 rats per group), i.e., the control group (Control): untreated rats, which were injected intraperitoneally (i.p.) with the vehicle; the model group (TAA): starting at week 3, rats were injected with 200 mg/kg TAA (twice/week, i.p.) for 8 weeks [2]; and the protective group (RES + TAA): rats received resveratrol suspension (20 mg/kg, orally) daily for 10 weeks and injected with 200 mg/kg TAA (twice/week, i.p.) from week 3 to week 10. At the end of the experiment, blood samples were obtained under anaesthesia, rats were culled, and liver tissue specimens were collected.

### 2.2. Histological Examination

H&E and Masson’s trichrome staining of tissue sections were used to examine liver tissue architecture and to quantify liver scarring, respectively [28]. Embedded liver sections (5 μm thick) were dewaxed, rehydrated, and stained with H&E and Masson’s trichrome stains.

### 2.3. Immunohistochemistry of p53 and α-Smooth Muscle Actin (α-SMA)

As previously reported [29], we performed the immunohistochemical staining after antigen retrieval by incubating tissue sections with the primary antibodies; anti-p53 antibody (Abcam, Cambridge, UK) or anti-α-SMA (Dako, Santa Clara, CA, USA) overnight at 4 °C and the secondary antibody was added for 30 min. Sections were counter stained with Meyer’s haematoxylin. The percentage of the area of collagen deposition in sections stained with Masson’s trichrome mentioned above, as well as the percentage of the areas of p53 and α-SMA immunostaining was morphometrically assessed using “Leica Qwin 500 C” image analyser (Cambridge, UK). The ANOVA and the post-hoc analysis (Tukey test) were used for comparing the quantitative data, which are presented as means ± standard deviations (SD). *p*-values < 0.05 were deemed statistically significant.

### 2.4. Quantitative Real-Time Polymerase Chain Reaction (qRT-PCR) of TIMP-1 and MiR-155

Total RNA was extracted from rats’ livers as previously described [1] and triplicate cDNA samples and standards were amplified with primers specific for TIMP-1 (sense, 5′-GGT TCC CTG GCA TAA TCT GA-3′; antisense, 5′-GTC ATC GAG ACC CCA AGG TA-3′) and β-actin as a housekeeping gene, whereas miR-155 was amplified as previously described (Aboulhoda et al., J Cell Physiol. 2021, 236:5994–6010) using primers specific for miR-155 (sense, 5′-CGCAGTTAATGCTAATTGTGATAG-3′; antisense, 5′-TCCAGTTTTTTTTTTTTTTTCAAGGT-3′), and the endogenous control suitable for miR-155, snU6RNA (sense, 5′- ATACAGAGAAGATTAGCATGGCC-3′; antisense, 5′- GTCCAGTTTTTTTTTTTTTTTCGAC-3′). The manufacturer’s software was used to determine the relative expression.

### 2.5. Evaluation of Biomarkers for Oxidative Stress, Antioxidants, and Hepatic Damage in the Liver Tissue and Blood

ELISA kits (Cayman Chemical, MI, USA) for the determination of liver malondialdehyde (MDA) and superoxide dismutase (SOD) were done according to the written instructions. Serum levels of high sensitivity C-reactive protein (hs-CRP, ASSAYPRO, Saint Charles, MO, USA), alanine aminotransferase (ALT), and aspartate aminotransferase (AST) (Randox Laboratories, Antrim, UK) were measured as per the manufacturer’s instructions.

### 2.6. Statistical and Morphometric Analysis

Data analyses were achieved using SPSS (version 25.0). One-way ANOVA was used for statistical comparisons, followed by Tukey’s post hoc test. Pearson’s correlation statistical analysis was applied to detect probable significance between two different groups. *p* ≤ 0.05 was considered statistically significant.

## 3. Results

### 3.1. Induction of MiR-155 and Oxidative Stress by TAA Intoxication Is Inhibited by Resveratrol

miR-155 induces ROS and liver fibrosis [16,17]. To investigate the ROS/p53 axis mediated liver fibrosis caused by TAA intoxication associated with miR-155 upregulation with and without RES incorporation, we first assessed liver tissue levels of miR-155, oxidative stress (MDA), and antioxidant (SOD) in all animals at week 10. TAA significantly (*p* < 0.0001) augmented miR-155 expression (Figure 1A), lipid peroxidation measured as MDA (Figure 1B) and ameliorated SOD (Figure 1C), which was completely protected by RES in (B), but still significant in (A and C) to the control group. In addition, liver injury biomarkers ALT and hs-CRP were evaluated in all animal groups. RES significantly (*p* ≤ 0.004) inhibited TAA-induced ALT (Figure 1D) and hs-CRP (Figure 1E). A positive correlation between miR-155 expression score and ROS (MDA) tissue levels (nmol/gm) was observed (r = 0.889; *p* < 0.0001) (Figure 1F).

### 3.2. Resveratrol Inhibits TAA-Induced Biomarker of Apoptosis p53 in Liver Tissues

In cell signalling, biomarker of apoptosis (p53) is located downstream of ROS [30]. The efficacy of resveratrol to suppress TAA-induced hepatic p53 in rats was investigated. Immunohistochemical analysis of p53 in liver sections of TAA-treated rats revealed a substantial increase in the expression of p53 protein (Figure 2B,D) compared to a weak p53 expression in the control group (Figure 2A), which were significantly (*p* < 0.0001) inhibited by resveratrol in the RES + TAA group (Figure 2C,D) to levels still significant (*p* = 0.0093) to the control group.

### 3.3. Resveratrol Is Associated with the Inhibition of Liver Injury and Fibrosis Induced by TAA

p53 is involved in the development of hepatic and cardiac fibrosis [31,32]. Therefore, in view of the results described above that showed the inhibition of TAA-induced miR-155/ROS/p53 axis by resveratrol at week 10, we assessed levels of liver injury and fibrosis as well as the profibrogenic biomarkers TIMP-1 and α-SMA with and without resveratrol incorporation (Figure 3). Compared to normal tissue architecture in the control untreated rats (Figure 3A) that shows cords of hepatocytes with acidophilic cytoplasm and vesicular nuclei surrounding portal tracts (P) separated by blood sinusoids, H&E-stained liver tissue sections of the model group (Figure 3B) showed liver tissue damage induced 8 weeks following TAA treatment, as demonstrated by disturbed parenchymatous architecture, dilated and congested central veins (V) and blood sinusoids (s), in addition to thick fibrous septa with inflammatory cell infiltration (not shown). The hepatocytes illustrate vacuolated cytoplasm and dark shrunken nuclei (arrowhead). Resveratrol treatment appeared to inhibit the deleterious effects of TAA (Figure 3C). However, some hepatocytes display cytoplasmic vacuolation (arrow). Furthermore, the control group’s liver sections stained with Masson’s trichrome revealed unremarkable collagen accumulation in the portal area with no inflammatory cells (Figure 3D), whereas stained liver sections of the experimental rats (TAA group) showed in the portal area and septum substantial deposition of course collagen (fibrosis), and infiltration of the portal tract with inflammatory cells (Figure 3E). Collagen deposition was completely inhibited with RES treatment (Figure 3F,G). Resveratrol also significantly (*p* < 0.0001) ameliorated TAA-induced TIMP-1 mRNA hepatic levels (Figure 4H) and α-SMA protein expression (Figure 3I).

### 3.4. Correlation between Liver Fibrosis Score and miR-155, Oxidative Stress, and Apoptosis

The correlation between liver fibrosis and the tissue levels of miR-155, ROS, p53, and TIMP-1 was determined. This is important when drawing an association between pathogenesis of TAA-induced liver scarring and these biomarkers and to ascertain that the use of resveratrol is beneficial in chronic liver disease induced by TAA and possibly by other causative agents. A significant (*p* < 0.0001) positive correlation between collagen score (% fibrosis) with miR-155 expression (Figure 4A), MDA (nmol/gm) (Figure 4B), p53 protein expression (Figure 4C), and TIMP-1 gene expression (Figure 4D) was observed.

## 4. Discussion

This study used immunohistochemical, molecular, and biochemical approaches to demonstrate an association between miR-155 and the ROS/p53 axis-mediated liver fibrosis induced with TAA in a rat model of chronic hepatic injury. Furthermore, we have been able to show that the natural antioxidant polyphenolic compound resveratrol is able to significantly protect against miR-155 elevation and the induction of the investigated axis of fibrosis as well as liver injury caused by the hepatotoxic chemical TAA (Figure 5). This further complements our report on the protective effect of the antioxidant metformin on TAA-induced liver damage via a different cell signalling axis of fibrosis, mammalian target of rapamycin (mTOR)-hypoxia-inducible factor-1-alpha (HIF-1α) [1].

In cell signalling, (i) miR-155 is located upstream of ROS, since miR-155 was reported to induce ROS generation through downregulation of antioxidation-related genes, such as Nfe2l2, Sod1, and Hmox1 in mesenchymal stem cells obtained from 18-month-old mice [17] and (ii) miR-155 promotes liver fibrosis and steatohepatitis, and profibrotic genes were inhibited in miR-155 knockout mice treated with hepatotoxic agents, such as alcohol and carbon tetrachloride [16]. miR-155 targeted genes were also involved in metabolism of lipid (Fab4 and cpt1a) and early fibrosis (C/EBPβ and Smad3) [33]. In addition, hepatic intoxication by TAA is well established [1,2], and pathological insults induced by this agent augment miR-155, ROS, p53, TIMP-1 as well as liver fibrosis (Figure 1, Figure 2 and Figure 3). Our data, therefore, are in agreement of previous reports that showed that (i) the induction of ROS/p53 axis is associated with cardiac fibrosis in rats [29]; (ii) ROS is located upstream of p53 [34] and p53 and ROS form a positive feedback loop, resulting in a vicious cycle that further provokes oxidative stress [35]; (iii) miR-155 augments ROS and apoptosis in human cerebral microvessel endothelial cells [36]; (iv) targeting oxidative stress (ROS) is suggested to treat liver fibrosis [37]; (v) p53 deletion prevents liver and cardiac fibrosis in mice [12,31] and deletion of the MDM2 gene specific for p53 degradation enhanced liver fibrosis in mice [12]; (vi) TIMP-1 promotes liver fibrosis as well as age-related renal fibrosis in transgenic mice with human TIMP-1 [14,38]; and (vii) resveratrol mitigated carbon tetrachloride-induced hepatic fibrosis in mice through interleukin-10 increase and downregulation of nitrosative stress [39].

In summary, we have demonstrated that pathological insults induced by TAA intoxication induce miR-155/ROS/p53 axis mediated liver fibrosis, which were otherwise protected for a period of 10 weeks with resveratrol in rats.

### Limitations of the Study

We demonstrated an association between miR-155 upregulation and liver fibrosis. However, to conclusively determine that miR-155 dysregulation is involved in the progression of hepatic fibrosis, we suggest a future study that examines the use of miR-155 knockout mice and the specific inhibitor, miR-155 siRNA that would provide more strength to the observed findings.

## Figures and Tables

**Figure 1 diagnostics-12-01762-f001:**
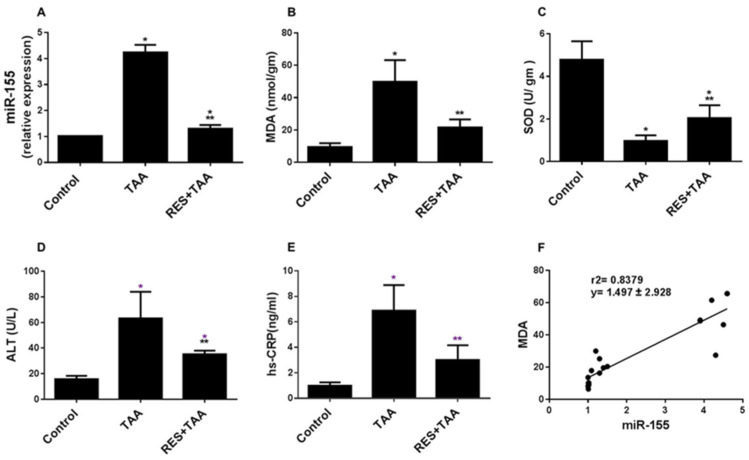
Resveratrol (RES) inhibits miR-155, ROS, and biomarkers of liver injury induced by TAA. Tissue (**A**–**C**) and blood (**D**,**E**) levels of miR-155 (**A**), MDA (**B**), SOD (**C**), ALT (**D**), and hs-CRP (**E**) were determined in all rat groups (Control, TAA, and RES + TAA) at week 10. (**F**) Correlation between miR-155 score and MDA. * *p* ≤ 0.046 versus control, ** *p* ≤ 0.023 versus TAA. TAA: thioacetamide; miR-155: microRNA-155; MDA: malondialdehyde; SOD: superoxide dismutase; ALT: alanine aminotransferase; hs-CRP: high sensitivity C-reactive protein.

**Figure 2 diagnostics-12-01762-f002:**
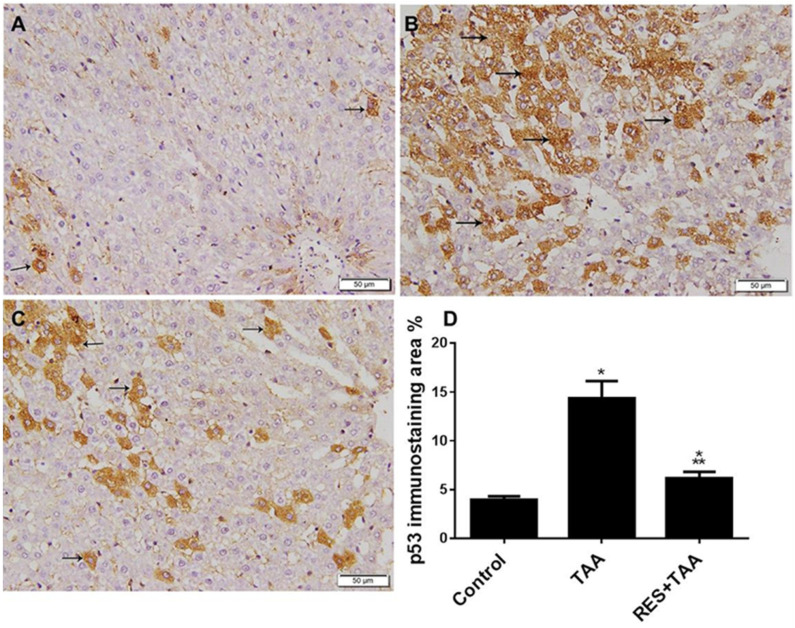
Resveratrol (RES) inhibits TAA-induced liver p53 expression. Immunohistochemistry of p53 protein expression in liver specimens (×200) of all rat groups was evaluated; Control (**A**), TAA (**B**), and RES + TAA (**C**) at week 10 from control (**A**), TAA (**B**), and RES + TAA (**C**) rat groups are illustrated. In (**A**,**C**), arrows point to the weak positive p53 immunostained cells, whereas arrows in (**B**) point to the strong positive p53 immunostained cells. The histograms in (**D**) discuss the quantitative analysis of p53 immunostaining from the above groups. * *p* < 0.01 versus control, ** *p* < 0.0001 versus TAA. p53: tumour suppressor p53; TAA: thioacetamide.

**Figure 3 diagnostics-12-01762-f003:**
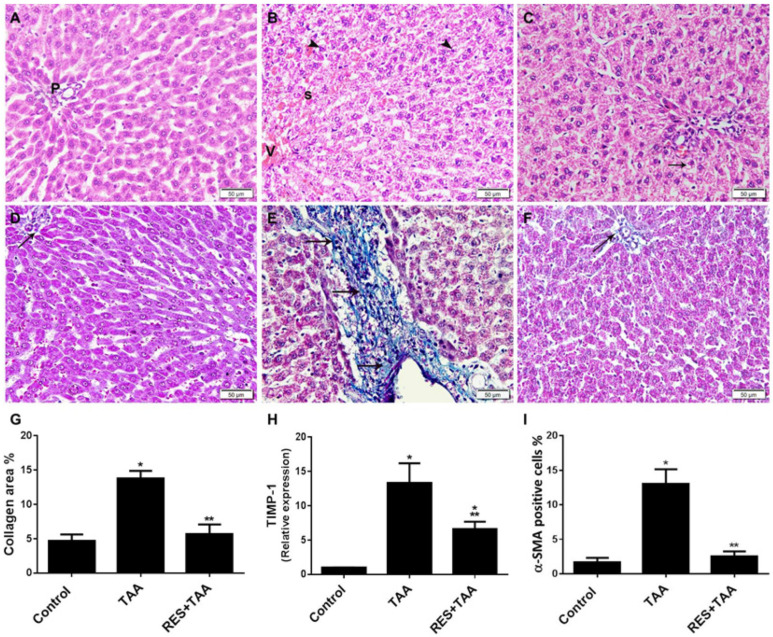
Induction of liver injury and fibrosis by TAA appears to be ameliorated by resveratrol (RES). (**A**–**C**) H&E-stained images (×200) of liver sections across all rat groups, i.e., control (**A**), TAA (**B**), and RES + TAA (**C**), are displayed. (**D**–**F**) Images (×200) of Masson’s trichrome stained liver sections across all rat groups, i.e., control (**D**), TAA (**E**), and RES + TAA (**F**), are displayed. In (**D**,**F**), the arrow points to the thin collagen fibres in the portal area, whereas in (**E**), arrows point to the thick collagen fibres deposit in the portal area and septum. The histograms in (**G**–**I**) provide a quantitative analysis of liver fibrosis calculated as the percentage of collagen deposition determined from Masson’s trichrome stain (**G**), TIMP-1 relative gene expression (**H**), and a-SMA protein expression (**I**). * *p* < 0.001 versus control, ** *p* < 0.0001versus TAA. TIMP-1: tissue inhibitor of metalloproteinases-1; TAA: thioacetamide; α-SMA: alpha smooth muscle actin.

**Figure 4 diagnostics-12-01762-f004:**
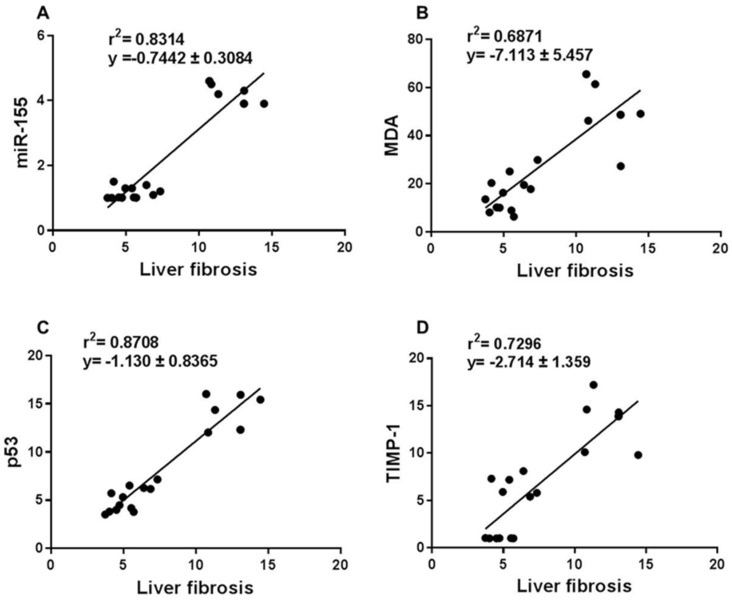
Correlation between liver fibrosis score and miR-155/ROS/p53 axis of fibrosis. Across all groups of rats, the above parameters were measured at the end of the study, at week 10, and the correlation between liver fibrosis score measured as the percentage of collagen deposition and miR-155, MDA, p53, and TIMP−1 are shown (**A**–**D**). miR-155: microRNA-155; MDA: malondialdehyde; p53: tumour suppressor p53; TIMP-1: tissue inhibitor of metalloproteinases-1; TAA: thioacetamide.

**Figure 5 diagnostics-12-01762-f005:**
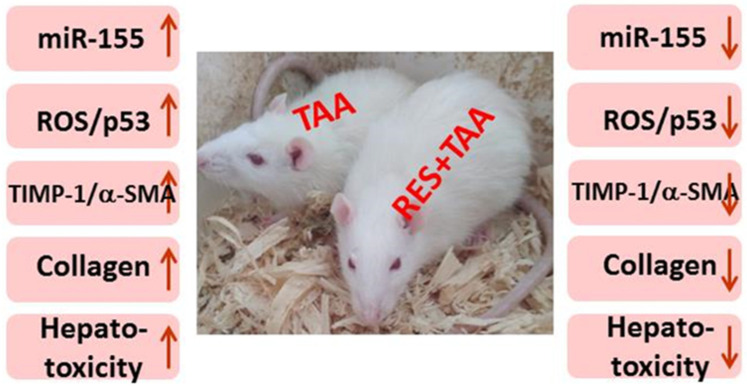
Proposed model for TAA induced miR-155/ROS/p53 axis mediated liver fibrosis appears protected by resveratrol. miR-155: microRNA-155; MDA: malondialdehyde; p53: tumour suppressor p53; TIMP-1: tissue inhibitor of metalloproteinases-1; α-SMA: alpha smooth muscle actin.

## Data Availability

The data that support the findings of this study are available on request from the corresponding author.

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
