# Peer review of "MiR-155 Dysregulation Is Associated with the Augmentation of ROS/p53 Axis of Fibrosis in Thioacetamide-Induced Hepatotoxicity and Is Protected by Resveratrol"

_diagnostics, 2022, doi:10.3390/diagnostics12071762_

Round 1
Reviewer 1 Report
Reviewer’s comments
This article primarily focuses on the role of miR-155 dysregulation in the process of hepatic fibrosis. This article has its own originality, and seems to be very interesting. However, the experimental designs are partially inappropriate. The explanation for the results obtained from this study seem to be insufficient. It is regrettable to say that this article is not acceptable for publication. Please refer to the comments shown below.
Major
#1. Experimental protocol in each group (control, TAA group and RES+TAA group) should be illustrated. The number of experimental animals in each group should be also addressed.
#2. An ethical guideline for the experimental animals in this study should be described.
#3. The histological evaluation of TAA-induced hepatotoxicity should be conducted using HE-stained liver specimens. After that, the efficacy by the treatment with resveratrol should be estimated.
#4. If the authors would like to demonstrate that the upregulation of miR-155 is involved in the progression to hepatic fibrosis, the evidence that downregulation of miR-155 by siRNA results in the improvement of hepatic fibrosis should be provided.
#5. The authors should document the target gene for miR-155. Also, the putative mechanism by which upregulation of miR-155 caused the development of hepatic fibrosis should be illustrated in more detail.
#6. The authors should describe several limitations in this study.
Minor
#1. The references can be abbreviated followed by the format. However, some of the references were not abbreviated.
#2. The statement on the areas % of collagen deposits (page 3, lines 122-124) should be moved to “Materials and Methods”.
#3. The units of X-axis and Y-axis in Figure 4A-4D should be addressed. Also, p-values for Pearson correlation statistical analyses should be addressed in Figure 1F and Figure 4A-4D.
#4. HIF-1a (page 7, line 211) should be spelled out.
Author Response
"Please see the attachment."

Reviewer 2 Report
In this manuscript, Dawood et al. reported the function of miR-155 is associated with hepatotoxicity mediated through the ROS/p53 signaling pathway. This is a novel discovery in terms of linking the role of miR-155 to liver pathology. This work used a rat model involving thioacetamide (TAA) injection with and without resveratrol (RES) treatment, a compound that produces a protective effect on the injury. This study was well-designed and had sufficient experimental and control groups. I have a few minor comments on this work:
1, In section 2.4, it was not clear to me how qRT-PCR was performed on miR-155. Regular qRT-PCR can be used to detect mRNAs; however, miRNAs are too short to be reverse-transcribed using regular poly-dT primers. More details are required since detecting miRNA accurately is essential for the downstream analysis.
2, Although Figure 4 shows strong correlations of miR-155 with liver fibrosis, the direct target of miR-155 remains elusive. Have the authors predicted its targets? This is important since other players in the pathway are all proteins, and miR-155 seems to be the only RNA modulator, implying that it may function in a different way.
3, The discussion section is short, which only summarizes the results, and mentions some previous works. I look forward to a more detailed discussion about: 1) What’s the impact of this work and how it can lead to the potential treatment of hepatotoxicity? 2) What are the unknown steps in this process which will need further work?
Overall, this work is suitable for publishing in CIMB, and I look forward to the revised manuscript.
Author Response
"Please see the attachment."
